# The Shepherd and the Hunter: A Genomic Comparison of Italian Dog Breeds

**DOI:** 10.3390/ani13152438

**Published:** 2023-07-27

**Authors:** Arianna Bionda, Matteo Cortellari, Luigi Liotta, Paola Crepaldi

**Affiliations:** 1Dipartimento di Scienze Agrarie e Ambientali—Produzione, Territorio, Agroenergia, University of Milano, Via Celoria 2, 20133 Milano, Italy; arianna.bionda@unimi.it (A.B.); paola.crepaldi@unimi.it (P.C.); 2Dipartimento di Scienze Veterinarie, University of Messina, Viale Palatucci 13, 98168 Messina, Italy; luigi.liotta@unime.it

**Keywords:** Italian dog breeds, shepherd dogs, hunting dogs, SNP, canine genomics

## Abstract

**Simple Summary:**

Shepherd and hunting dog breeds have distinct physical and behavioural characteristics due to their specialized roles. To understand the genetic basis of these differences, we compared the genomes of various Italian shepherd and hunting dog breeds. Our study involved analysing SNP data from 116 hunting dogs (representing 6 breeds) and 158 shepherd dogs (representing 9 breeds). We examined the genetic makeup, population structure, and relationships of these breeds and compared shepherd and hunting dogs with three complementary techniques. The results clearly showed that there are significant genetic differences between shepherd and hunting dogs. Specific regions of the genome were identified, containing genes associated with domestication, behavioural traits such as aggressiveness and gregariousness, and physical attributes including size, coat colour, and texture. This research provides valuable insights into the genetic factors contributing to the diverse characteristics observed in Italian hunting and shepherd dogs. Further investigation is warranted to explore the implications of these findings for dog health and breeding practices.

**Abstract:**

Shepherd and hunting dogs have undergone divergent selection for specific tasks, resulting in distinct phenotypic and behavioural differences. Italy is home to numerous recognized and unrecognized breeds of both types, providing an opportunity to compare them genomically. In this study, we analysed SNP data obtained from the CanineHD BeadChip, encompassing 116 hunting dogs (representing 6 breeds) and 158 shepherd dogs (representing 9 breeds). We explored the population structure, genomic background, and phylogenetic relationships among the breeds. To compare the two groups, we employed three complementary methods for selection signature detection: F_ST_, XP-EHH, and ROH. Our results reveal a clear differentiation between shepherd and hunting dogs as well as between gun dogs vs. hounds and guardian vs. herding shepherd dogs. The genomic regions distinguishing these groups harbour several genes associated with domestication and behavioural traits, including gregariousness (WBSRC17) and aggressiveness (CDH12 and HTT). Additionally, genes related to morphology, such as size and coat colour (ASIP and TYRP1) and texture (RSPO2), were identified. This comparative genomic analysis sheds light on the genetic underpinnings of the phenotypic and behavioural variations observed in Italian hunting and shepherd dogs.

## 1. Introduction

According to archaeological data, it is believed that dogs were domesticated between 15,000 and 10,000 years ago and probably used as hunting aids [1,2]. The partnership between humans and dogs had such a strong influence on human hunters that it led them to develop a new hunting style: instead of killing animals with axes and spears, they started to hunt using long-distance arrows and were aided in this process by dogs that tracked down, cornered, and transported prey back to the human hunters [3]. In a subsequent stage, after the shift to agriculture and farming, some dogs assumed a new role, helping in the management of domesticated livestock [4].

It is only from samples dating to the period around 3000–4000 years ago that we can find evidence of the presence of distinctive dog breeds [5]; this is when the selection of dogs became more and more directed toward specific functionalities. In fact, during the Roman period, several authors described different classes of dogs, including their physical and behavioural traits. For example, Columella in “*De re rustica*” distinguishes a dog meant to guard farmyards (*villaticus*, black-coated, squarely built, and with a large head) or livestock (*pastoralis*, white-coated, strong but quick), and a hunting dog (*venaticus*). Various *Cynegetica* further described the latter, which could specialize in scenting prey (*sagaces*), chasing it (*celeres*), or fighting it (*pugnax*, also used as war dog) [6,7].

Today, hunting dogs are divided into hounds and gun dogs. Hounds can base their hunting on their sight (sight hounds) or their smell (scent hounds). The first ones have been bred since antiquity to chase small game, whilst the latter were bred to track prey by following its trail. Alternatively, gun dogs made their appearance around the XIV century; they were developed to locate game and point to its location (pointing dogs), flush it out and drive it from its hiding spot for capture (i.e., flushing dogs, like spaniels), or collect it and return it to the hunter without damaging it (retrieving dogs) [1,8]. In contrast, shepherd dogs can be characterized as livestock guardians, which constantly watch and protect a flock from wild predators such as wolves and bears, and herding dogs, which, alternatively, cooperate with the shepherd to drive the livestock and keep it in a group.

At present, over 350 dog breeds are internationally recognized by the *Federation Cynologique Internationale* (FCI), including over 60 shepherd dog breeds and over 200 hunting dogs. Italy has 14 breeds (2 of which are shepherd dogs and 5 are hunting dogs) that are internationally recognized on a definitive basis; however, there are several local canine populations that have not yet obtained international acknowledgement as breeds but are still bred and used in practice by farmers and hunters. The selection of these dogs has been carried out for centuries, and it has mainly been based on their behavioural and working characteristics. For this reason, studying these populations from a genetic point of view could give some insight not only into their history [9,10,11,12,13,14,15] but also into the genes and genetic pathways that are most involved in the development of the particularities of distinct groups of dogs [16,17,18,19,20]. Some authors have investigated the selection signatures associated with specific behavioural [21,22,23,24,25,26,27,28,29] or morphological [2,16,24,30,31,32,33,34,35,36] phenotypes or breed groups [20,37,38], but, to the best of our knowledge, none have focused on the comparison of hunting and shepherd populations with a common cradle.

Therefore, the present study sought to compare genetic data, obtained using a high-density SNPchip, of 16 Italian dog breeds in order to investigate their genetic distances and the genomic differences between Italian shepherd and hunting dog breeds.

## 2. Materials and Methods

### 2.1. Animals and Genotyping

Data analysed in this study were acquired through the genotyping of 274 dogs belonging to 15 different Italian dog breeds using the CanineHD BeadChip (Illumina, San Diego, CA, USA) [10,12,17,39,40]. The “Hunting dogs” group (HD) included 12 Bracco italiano (BRAC), 24 Cirneco dell’Etna (CIRN), 24 Lagotto Romagnolo (LAGO), 16 Segugio Italiano Pelo Forte (SIPF), 16 Segugio Italiano Pelo Raso (SIPR), and 24 Spinone Italiano (SPIN); the “Shepherd dogs” group (SD) consisted of 19 Pastore Apuano (APUA), 15 Bergamasco shepherd dogs (BERG), 15 Pastore d’Oropa (DORO), 30 Fonni’s dogs (FONN), 23 Lupino del Gigante (LUGI), 12 Mannara dogs (MANN), 20 Maremma and the Abruzzi sheepdogs (MARM), 10 Pastore della Lessinia e del Lagorai (PALA), and 14 Pastore della Sila (SILA). Information obtained from breed standards and pictures of these populations are reported in Appendix A. In order to derive geographical and functional outgroups, other non-Italian breeds were included, namely, 10 Bloodhounds (BLDH, scent hounds), 7 Caucasian shepherd dogs (CAUC, livestock guardians), 10 German shepherd dogs (GSD, herding dogs), 10 German shepherd short-haired pointers (GSHP, gun dogs), and 10 Mastiffs (MAST, molossers-type property guardian).

Given the peculiar population structure of purebred dogs, even a small number of carefully chosen subjects can serve as representative samples of their respective breeds [41]; indeed, the sample size per breed analysed in the present study is consistent with that of other comparable genomic studies [10,15,42].

The initial dataset underwent quality control to remove all the individuals and SNPs with a call rate < 95% and SNPs with a minor allele frequency (MAF) < 1% or located on sex chromosomes. Moreover, only dogs that were not directly related were retained. These operations were performed using PLINK 1.9 software [43].

Lastly, genotype data were phased using BEAGLE 4.1 software.

### 2.2. Genomic Analyses

The genomic population structures of all the subjects were investigated by performing a multidimensional scaling analysis (MDS) of their genomic distances with PLINK, and an admixture analysis was carried out with ADMIXTURE v1.3.0 [44]. Reynolds genetic distances among breeds were calculated and used for creating a phylogenetic tree with an in-house script.

In order to establish numerically balanced groups, breed sample size was reduced to a maximum of 24 dogs for the analyses of selection signatures.

Genomic regions under selection in HD and SD were detected by both comparing the two groups using Wright’s fixation index (F_ST_) and the Cross Population Extended Haplotype Homozygosity (XP-EHH) analyses and searching for genomic regions with the highest level of homozygosity within each group. Estimates of F_ST_ allow for the detection of regions with the highest variance in allele frequency between two populations [45], while XP-EHH is a haplotype-based approach that evaluates the differences in the length of extended haplotypes between two populations [46].

The F_ST_ statistic for each SNP was calculated using PLINK, and the top 1% SNPs were considered to be those that best differentiated the two groups. XP-EHH values were assessed using Selscan 1.1.0 software [47] and then normalized; the SNPs falling in the top 1% of the empirical distribution of XP-EHH values were further considered. The SNPs that were identified through both analyses were mapped to CanFam3.1, and the genes contained in the regions where they located were studied.

Runs of homozygosity (ROH) are continuous stretches of homozygous segments inherited from a common ancestor. If a locus is under selective pressure, the regions around it show decreased haplotype diversity and increased homozygosity; thus, regions with a high frequency of ROH can be used to identify selection signatures of a population [48]. ROH were calculated for each dog using the *--homozyg* function of PLINK, which relies on a scanning window approach. A ROH was called using the following parameters: *--homozyg-window-snp* 50, *--homozyg-window-het* 0, *--homozyg-window-missing* 5, --homozyg-density 50, --homozyg-gap 100, --homozyg-kb 1000, and --homozyg-snp 50. The ratio of the number of times a SNP appears in a ROH to the size of each group (H-score) was calculated. The markers in the top 1% of H-scores were retained, and associated genes were further investigated.

### 2.3. Further Breed Comparison

In order to further investigate the differences between similar breeds, we performed an F_ST_ analysis comparing BRAC vs. SPIN and SIPR vs. SIPF. In particular, these breeds present similar morphologies but different coat textures, with SPIN and SIPF being characterised by rough coats and furnishing. The SNPs in the top 1% of F_ST_ values were mapped using CanFam3.1.

## 3. Results

### 3.1. Population Structure

After the exclusion of dogs with low-quality data and those that were directly related, the dataset used for the population structure analyses comprised a total of 255 Italian dogs, 47 non-Italian dogs, and 120,828 SNPs.

Figure 1 graphically represents the MDS results. Overall, all the breeds are quite distinguishable, with the exception of SIPR-SIPF and most of the livestock guardian populations. Along the horizontal axis, corresponding to the first principal component that explains 4.42% of the population’s genomic variability, the dogs are clearly separated according to the functional group they belong to. Specifically, the two pointing breeds (BRAC and SPIN) have the most negative values, followed by the analysed hound dogs (LAGO, SIPR, SIPF, and CIRN); instead, the lowest and highest positive values are associated with livestock guardian (MARM, MANN, FONN, and SILA) and herding shepherd dogs (BERG, PALA, APUA, DORO, and LUGI), respectively. The second principal component (3.60%) mainly isolates the LAGO breed, which is probably due to the fact that this breed is currently mainly used for truffle searching instead of hunting. The third principal component (3.3%) separates CIRN, which are sighthounds, from SIPR and SIPF, which are scent hounds. A similar disposition (although with inverted axes) was observed when non-Italian breeds were included (Appendix A): GSD was located at the extreme left of herding dogs; CAUC and GSHP clustered with livestock guardians and gun dogs, respectively; BLDH was located between hounds and gun dogs along the horizontal axis but was separated from all the other dogs along the vertical second component; and MAST was near the livestock guardians and the hounds.

The dendrogram based on Reynolds distances (Figure 2) confirms previous results, showing a clear grouping of herding dogs and livestock guardians on the left and gun dogs and hounds on the right. When non-Italian breeds were taken into consideration, we observed that GSHP clustered with other gun dogs and that GSD was near herding shepherd dogs, whereas BLDH, MAST, and CAUC clustered together between livestock guardians and herding dogs (Appendix A).

With respect to the admixture analysis (Figure 3), the optimal number of structural clusters (K), equal to 11, was identified as the one with the lowest cross-validation value (CV). Using this model, the only breeds not characterised by a unique genetic signature are APUA, MANN, PALA, SIPR, and SIPF. While the sheepdogs show a very admixted background, the two Segugio Italiano breeds are very similar to each other but different from all the other populations. It is also worth noticing that when only two clusters are accounted for in the model, the distinction between shepherd and hunting dogs is rather clear.

### 3.2. Selection Signatures

In order to balance the number of subjects for each breed, a maximum of 24 subjects per breed were retained, leading to the exclusion of six FONN.

There were 350 SNPs in the top 1% of the normalized XP-EHH values (2.78–6.55), and the regions they were located in contained 120 different genes. Alternatively, the top 1% of F_ST_ (0.17–0.63) consisted of 408 SNPs associated with 266 genes. The two analyses shared 71 SNPs located on 40 different genes, which are reported in Table 1.

In shepherd dogs, the top 1% H-score corresponded to SNPs located on 18 ROH islands on 12 chromosomes, whereas in hunting dogs, it included 22 ROH islands on 11 chromosomes. The two groups partially shared eight ROH islands, which were located on chromosomes 5, 6, 13, 22, 25, 30, and 34 (Table 2). We identified SNPs on a total of 44 genes that were located in ROH regions identified in both groups, while 136 genes and 88 genes were only found in shepherd and hunting dogs, respectively, as presented in Appendix A and Figure 4.

### 3.3. Breed Comparison

As shown in Figure 5, through the comparison of BRAC and SPIN and SIPF and SIPR, the highest F_ST_ values were found on CFA 13. In particular, eight SNPs comprised between 8,200,350 and 8,401,561 bp had an F_ST_ value of between 0.47 and 0.63 in the SIPF-SIPR comparison, whereas eleven SNPs located between 8,809,543 and 9,455,586 bp had an F_ST_ value ranging from 0.86 to 0.97 when comparing BRAC and SPIN, but they were not included in a genic region.

Appendix A reports the results of these analyses, including all the genes that harboured the top 1% SNPs. It should be noted that in both the comparisons, the RSPO2 gene was included among the most-differentiating genes.

## 4. Discussion

This study investigated the genomic differences in Italian hunting and shepherd dog breeds. Population structure analyses, including multidimensional scaling, genomic distances, and admixture analyses, showed a clear distinction between the two groups but also between gun dogs and hounds on one side and livestock guardian and herding shepherd dogs on the other, as previously reported [40].

To identify the selection signatures that differentiate hunting and shepherd breeds, we employed three complementary analyses: F_ST_, XP-EHH, and ROH. Some of the genomic regions identified in our study contain genes have been previously reported to differentiate different dog functional groups. For instance, the gene JAK2 was found to be associated with predatory behaviour in a comparison of hounds and herding dogs [23]. In addition, HTT and CDH12, along with four other genes found in hunting dogs’ ROHs (GRK4, MSANTD1, RGS12, and ZFYVE28), were found to be positively selected in sport-hunting dogs [37], and several genes included in the ROHs of both the groups (CAB39L, CDADC1, CYSLTR2, FNDC3A, PHF11, RCBTB1, RCBTB2, SETDB2, and EIF3E) or hunting dogs only (WBSCR17 and RSPO2, which were also found through F_ST_ and XP-EHH analyses, and TMEM74) were included in a study about the genomics of canine pointing behaviour [20]. Furthermore, our results partially overlap with those of a recent study that described different dog lineages at the genomic level. For example, the genes contained in the hunting dogs’ ROH on CFA 3 were identified as characteristic of spaniel and pointer lineages [19].

These two groups of dogs have been historically selected for completely different tasks that require specific behavioural traits, abilities, and aptitudes. Thus, it is not surprising that a number of the regions detected in the present study harbour genes that have been reported to be associated with dog behaviour. Of particular interest is the WBSCR17 gene, which was identified in all the performed analyses. Indeed, this gene appears to be differently selected in dogs compared to wolves and a good predictor for success in assistance-dog training programs, which is possibly due to its role in the development of gregariousness, as seen in Williams–Beuren syndrome in humans [38,42]. Some other genes have been linked to dog domestication, such as RNF103, identified via F_ST_ and XP-EHH analyses, and a long ROH belonging to the shepherd dog group, which was determined to be differently selected in tame and aggressive foxes in Belyaev’s farm-fox experiment [49]. Other genes are related to aggressiveness, such as CADM1 and HTT, which seem to be related to fighting behaviour and resident–intruder aggressiveness in mice [50]; TMEM74 and TRHR, which are present in hunting dogs’ ROHs and associated with dogs’ aggressiveness toward conspecifics [51]; and HTR1F, which is contained in a ROH highly represented in shepherd dogs and is involved in the serotonine pathway and associated with canine aggression [28]. These differences can be related to the fact that hunting dogs, especially hounds, usually work in groups, whereas livestock guardians are tasked with defending a territory and herds from animal or human intruders [8,19,52]. Another crucial aspect of the human–dog relationship is communication, which differs depending on the task required of the dog. Among the genes that differentiate the two studied groups, SDCCAG8 has been previously linked to dogs’ response to human pointing, UGGT1 has been linked to physical reasoning, and VAV3 as well as the hunting dog ROH’s ZFYVE28 have been linked to inhibitory control [27,53].

There are other aspects that have been favoured differently in dogs according to their intended task. For example, Kim et al. (2019) reported that sport-hunting dogs might have been selected for the inhibition of their hearing function in order to reduce these dogs’ startle response during hunting activities [37]. Among the genes identified via F_ST_ and XP-EHH analyses, four are actually associated with hearing loss: CADM1 [54], ESRRB [55,56], AKT3 [57,58], and ATP2B2 [59,60].

While most of the shepherd dog breeds included in this study can exhibit a variety of coat colours and textures, these aspects are much more standardized in hunting dogs. One of the consensus genes distinguishing the two dog groups is TYRP1, whose recessive variants lead to altered eumelanin production and the consequent appearance of a brown/liver/chocolate coat colour [61]. According to the standard of the breeds, the only shepherd dogs that can have brown coats are the Mannara dog and the Pastore della Lessinia e del Lagorai, whereas among the hunting dogs, only the two Segugio Italiano can only have black noses, indicating the absence of the homozygous recessive variant. The ASIP gene is another important factor of coat colouration [62] and was included in hunting dogs’ ROHs in this study. This finding might have been due to the limited number of admitted colours in these breeds: with the exception of Lagotto Romagnolo and the Segugio Italiano breeds, which are allowed to present a black and tan coat, the standards require a uniform colour. However, in some cases, such as in the Cirneco dell’Etna, the absence of eumelanin can depend on the Extension (E) locus (controlled by the MC1R gene), which has a possible epistatic effect on the ASIP genotype [63,64,65]. Another gene found through both F_ST_ and XP-EHH and in hunting dogs’ ROHs is RSPO2. This gene is responsible for the presence of facial and leg furnishing on dogs and, along with the FGF5 gene, plays a role in the determination of hair length [17,66]. With the exception of the Lagotto Romagnolo, all the enrolled hunting breeds are short-haired, and two of them, namely, the Segugio Italiano a Pelo Forte and the Spinone Italiano, present furnishing. In contrast, shepherd dog breeds are all long-haired, even though short-haired varieties of the Lupino del Gigante, Fonni’s dog, and Pastore d’Oropa breeds exist. In this respect, we also compared two hunting dog breeds that present rough coats and furnishing (Spinone Italiano and Segugio Italiano a Pelo Forte) with two other breeds very similar in their morphology and behaviour but with smooth coats (Bracco Italiano and Segugio Italiano a Pelo Raso, respectively). Interestingly, the very highest F_ST_ values were associated with SNPs located on CFA 13, very close to, but not overlapping, the RSPO2 gene. However, SNPs located within this gene were among the top 1% values for both the comparisons, supporting the association between RSPO2 and this coat texture. Moreover, several genes detected from the comparison between the Bracco Italiano and Spinone Italiano breeds were found to be associated with fur texture (EPHA7 and EX1) or length (ANTXR2 and GPAT3) in a study on dogs [51], whereas PRLR’s mutations are responsible for slick hair in cattle (OMIA:001372-9913) and Z-linked feathering in chickens and turkeys (OMIA:000380-9031; OMIA:000380-9103). Similarly, ANGPT1, TRPS1, and EPHA7, which the aforementioned study [51] associated with fur texture, distinguished the two Segugio Italiano breeds, and so did KRT71, whose mutations induce the appearance of a curly coat in dogs (OMIA:000245-9615) and some cat breeds (OMIA:001581-9685 and OMIA:001712-9685) and hypotrichosis in cats and cattle (OMIA:001583-9685 and OMIA:002114-9913). It is worth mentioning that other genes that differentiated Spinone Italiano and Bracco Italiano are instead related to reproductive function; in particular, SNX29 was previously associated with litter size in goats [67]. However, according to the litter-birth reporting forms available on the Breed Club websites (www.ilbraccoitaliano.org/wp/cucciolate/ (accessed on 20 June 2023) and www.spinone-italiano.it/cucciolate/ (accessed on 20 June 2023)), there is no significant difference between the two breeds’ litter sizes (on average, ±S.D., 7.4 ± 2.2 puppies for Bracco Italiano in 32 registered litters between 2021 and 2023, and 7.6 ± 2.8 puppies for the 20 Spinone Italiano litters registered in 2022 and 2023). SNX29 is also related to the growth trait in goats [68], whereas GAPVD1, KHDRBS3, LRRIQ3, and TENM3 might influence size in dogs [51,69]; however, the two breeds have minimal differences in terms of weight and height at withers according to their respective standards. Lastly, mutations in several of the identified genes can cause diseases in dog species, including metabolic (OMIA:002250-9615 and OMIA:000626-9615), ocular (OMIA:002179-9615, OMIA:002536-9615, and OMIA:000626-9615), ciliary (OMIA:001540-9615 and OMIA:002148-9615), and neurological diseases (OMIA:002240-9615, OMIA:002120-9615, and OMIA:001867-9615). In particular, the FYCO1-related cataract was identified in the Wirehaired Pointing Griffon [70], a breed that is very similar to the Spinone Italiano. It is important to underline that these findings do not provide any evidence regarding the presence of a particular disease in a breed but only regarding a difference in the allele frequency of those regions in the compared dogs. Nevertheless, these discoveries hold promise for future research, particularly given the current lack of reported data regarding the predisposition of these breeds to these diseases. Accordingly, it is crucial for breeders and veterinarians to report any cases they encounter, thereby facilitating in-depth studies and, ultimately, the better management and conservation of these breeds.

Returning to the findings of the comparison between the hunting and shepherd dog groups, we also found a number of genes contained in the hunting dogs’ ROH that have been associated with dog size. Among these, GHR [24,30,35,36,71], which encodes the receptor for the growth hormone, and LCORL [30,35,69,71,72] stand out. It should be considered that most of the investigated Italian hunting breeds are medium-sized, while the shepherd group is more heterogeneous, including very large guardian shepherd dogs and medium-sized herding dogs. Therefore, the presence of these ROH in the analysed hunting dogs could be related to the fact that size is an important selection criterion and has been highly standardized.

Lastly, it is worth reporting that one of the hunting dogs’ ROH (present in approximately half of the Lagotto Romagnolo and one third of Bracco Italiano, Cirneco dell’Etna, and Spinone Italiano but also Maremma and the Abruzzi sheepdogs and Bergamasco shepherd dogs) included the GDNF gene, whose recessive mutation has been proven to cause acral mutilation syndrome in some pointer and spaniel breeds [73]. This sensory autonomic neuropathy is associated with insensitivity to pain, which often leads to behaviours ranging from intense liking and biting to self-mutilation of the paws. It is important to note that no conclusive assumptions can be made based on the present dataset of dogs, as phenotypic information related to this issue is not available. However, this finding could be a valuable indication for future research on the subject in relation to Italian dog breeds.

## 5. Conclusions

Since antiquity, classifying dog breeds has always been a subject that has attracted a great deal of human attention. This interest can be traced from the ancient Romans’ first functional classification of dogs (*venatici*, *pastoralis*, and *villatici*) to Pierre Megnin’s 1896 classification based on head morphology (lupoid, braccoid, molossoid, and graiod). The establishment of the FCI in 1911 further categorized dog breeds into 10 groups, considering both morphology and function.

Based on this historical context, this scientific research work provides valuable and objective insights into the genomic diversity existing between Italian hunting and shepherd dogs. It reveals specific genomic regions that distinguish these two groups of dogs, particularly in terms of behaviour and morphology, such as size and coat characteristics. These findings underscore the significant role of genomic information in differentiating breeds intended for specific tasks. They also highlight the limitations of exclusively relying on phenotypic assessments, which may prove inadequate in certain cases.

## Figures and Tables

**Figure 1 animals-13-02438-f001:**
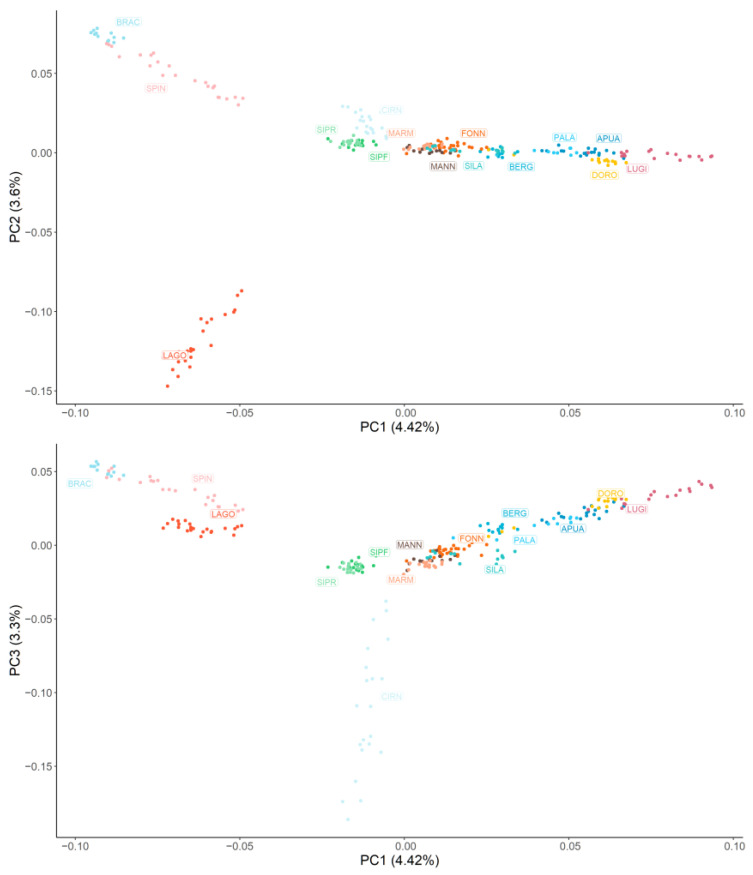
Graphical representation of the first three principal components (PC) of the multidimensional scaling analysis. Each dog is represented by a dot, and each breed is represented by a colour. Hunting dogs: Bracco italiano (BRAC), Cirneco dell’Etna (CIRN), Lagotto Romagnolo (LAGO), Segugio Italiano Pelo Forte (SIPF), Segugio Italiano Pelo Raso (SIPR), and Spinone Italiano (SPIN). Shepherd dogs: Pastore Apuano (APUA), Bergamasco shepherd dogs (BERG), Pastore d’Oropa (DORO), Fonni’s dogs (FONN), Lupino del Gigante (LUGI), Mannara dogs (MANN), Maremma and the Abruzzi sheepdogs (MARM), Pastore della Lessinia e del Lagorai (PALA), and Pastore della Sila (SILA).

**Figure 2 animals-13-02438-f002:**
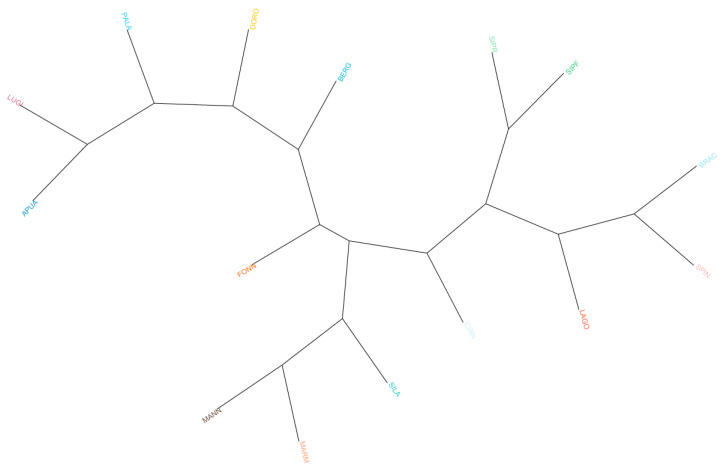
Phylogenetic tree based on Reynolds distances. Hunting dogs: Bracco italiano (BRAC), Cirneco dell’Etna (CIRN), Lagotto Romagnolo (LAGO), Segugio Italiano Pelo Forte (SIPF), Segugio Italiano Pelo Raso (SIPR), and Spinone Italiano (SPIN). Shepherd dogs: Pastore Apuano (APUA), Bergamasco shepherd dogs (BERG), Pastore d’Oropa (DORO), Fonni’s dogs (FONN), Lupino del Gigante (LUGI), Mannara dogs (MANN), Maremma and the Abruzzi sheepdogs (MARM), Pastore della Lessinia e del Lagorai (PALA), and Pastore della Sila (SILA).

**Figure 3 animals-13-02438-f003:**
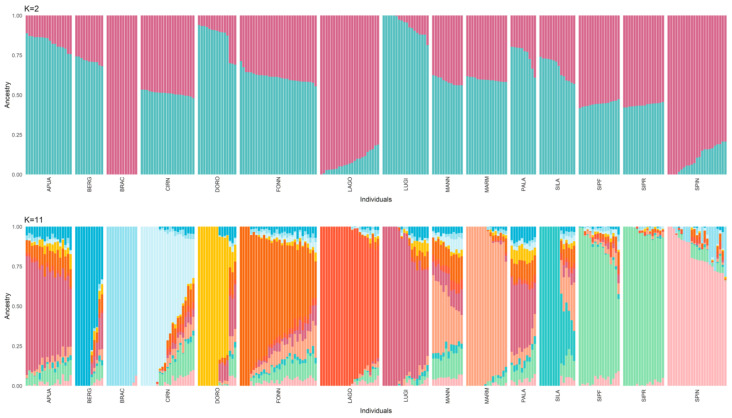
Admixture analysis considering 2 and 11 (identified as the best-fitting model) clusters (K). Each cluster is represented by a different colour. Hunting dogs: Bracco italiano (BRAC), Cirneco dell’Etna (CIRN), Lagotto Romagnolo (LAGO), Segugio Italiano Pelo Forte (SIPF), Segugio Italiano Pelo Raso (SIPR), and Spinone Italiano (SPIN). Shepherd dogs: Pastore Apuano (APUA), Bergamasco shepherd dogs (BERG), Pastore d’Oropa (DORO), Fonni’s dogs (FONN), Lupino del Gigante (LUGI), Mannara dogs (MANN), Maremma and the Abruzzi sheepdogs (MARM), Pastore della Lessinia e del Lagorai (PALA), and Pastore della Sila (SILA).

**Figure 4 animals-13-02438-f004:**
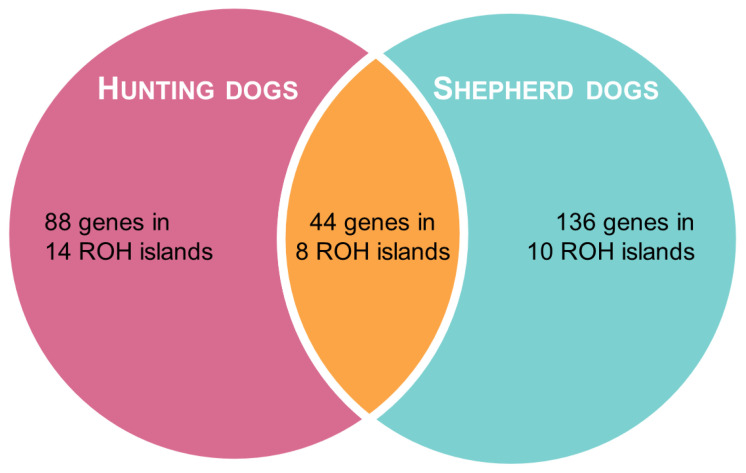
Venn diagram representing the results of ROH analysis.

**Figure 5 animals-13-02438-f005:**
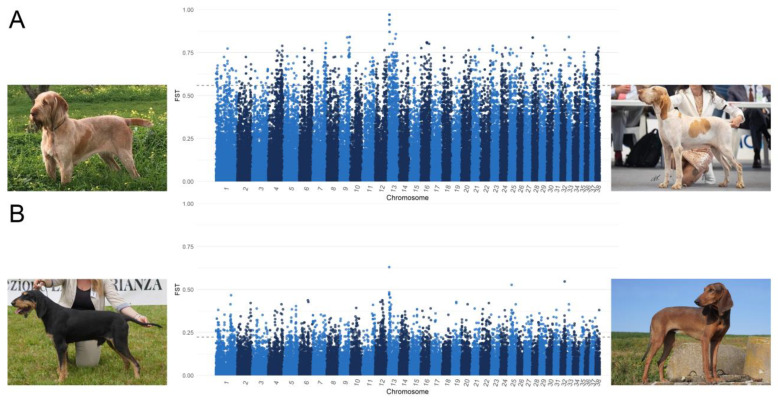
Manhattan plots of F_ST_ analysis comparing (**A**) Spinone Italiano (SPIN) vs. Bracco Italiano (BRAC) and (**B**) Segugio Italiano a Pelo Forte (SIPF) vs. Segugio Italiano a Pelo Raso (SIPR).

**Table 1 animals-13-02438-t001:** Genes harbouring the SNPs identified via both F_ST_ and XP-EHH analyses as the most differentiated hunting and shepherd dogs.

Gene Symbol	CFA	Gene Position	Gene Name
JAK2	1	93,321,921–93,435,774	Janus kinase 2
SYK	1	95,903,325–95,986,549	Spleen associated tyrosine kinase
ZNF599	1	117,643,042–117,656,167	Zinc finger protein 599
WDR41	3	29,073,170–29,286,566	WD repeat domain 41
UBE3A	3	35,376,364–35,440,245	Ubiquitin protein ligase E3A
HTT	3	61,080,904–61,221,554	Huntingtin
TMEM254	4	29,317,165–29,323,691	Transmembrane protein 254
C5orf42	4	71,430,719–71,566,921	Chromosome 5 open reading frame 42
CDH12	4	82,716,681–83,126,350	Cadherin 12
CADM1	5	17,865,290–18,191,185	Cell adhesion molecule 1
MMP20	5	29,106,717–29,154,396	Matrix metallopeptidase 20
KLHDC4	5	65,386,519–65,443,427	Kelch domain containing 4
WBSCR17	6	2,132,919–2,563,654	Williams–Beuren syndrome chromosome region 17
VAV3	6	43,729,781–44,000,437	Vav guanine nucleotide exchange factor 3
ADGRL2	6	65,596,760–65,786,099	Adhesion G protein-coupled receptor L2
SDCCAG8	7	34,396,345–34,631,157	Serologically defined colon cancer antigen 8
AKT3	7	34,636,388–34,931,586	AKT serine/threonine kinase 3
RHOJ	8	37,802,907–37,880,272	Ras homolog family member J
PPP1R36	8	38,975,044–39,006,998	Protein phosphatase 1 regulatory subunit 36
ESRRB	8	49,253,497–49,424,763	Estrogen related receptor beta
RPH3AL	9	45,333,293–45,420,867	Rabphilin 3A-like (without C2 domains)
TSPAN8	10	12,785,741–12,817,966	Tetraspanin 8
TRHDE	10	13,746,701–14,118,059	Thyrotropin releasing hormone degrading enzyme
TYRP1	11	33,317,645–33,335,498	Tyrosinase related protein 1
CLTA	11	52,659,891–52,680,985	Clathrin light chain A
RSPO2	13	8,610,233–8,755,897	R-spondin 2
CEP83	15	34,418,139–34,499,159	Centrosomal protein 83
TMCC3	15	34,629,079–34,850,528	Transmembrane and coiled-coil domain family 3
RNF103	17	38,549,288–38,570,900	Ring finger protein 103
EXT2	18	44,983,793–45,129,935	Exostosin glycosyltransferase 2
UGGT1	19	22,419,831–22,533,441	UDP-glucose glycoprotein glucosyltransferase 1
ATP2B2	20	7,863,328–8,034,474	Atpase plasma membrane Ca2+ transporting 2
SRGAP3	20	8,894,162–9,142,066	SLIT-ROBO Rho gtpase activating protein 3
GRM5	21	10,982,218–11,492,456	Glutamate metabotropic receptor 5
CPNE4	23	28,621,367–29,077,642	Copine 4
RPLP0	26	16,148,467–16,153,614	Ribosomal protein lateral stalk subunit P0
TDRD1	28	25,027,863–25,081,833	Tudor domain containing 1
ATRNL1	28	25,816,300–26,605,650	Attractin like 1
BMP3	32	5,207,833–5,231,966	Bone morphogenetic protein 3
DGKG	34	18,823,690–19,016,190	Diacylglycerol kinase gamma

**Table 2 animals-13-02438-t002:** Runs of homozygosity islands found in shepherd and hunting dogs.

CFA	Shepherd Dogs	Hunting Dogs
1	60,722,335–62,055,218	
2		75,050,435–75,093,914
3	1,181,619–1,332,812	
4		60,764,954–62,083,754
		90,826,390–91,354,784
		66,913,735–68,448,466
		70,219,838–70,306,623
		70,975,962–71,689,191
		73,608,702–74,243,079
		81,342,622–82,298,383
		84,651,292–86,449,303
5	2,035,658–3,736,188	2,394,506–2,498,212
6		1,091,264–2,670,297
	3,107,405–4,017,962	3,061,041–3,846,530
		35,840,074–36,402,977
13	991,157–2,333,650	
	3,220,205–4,326,026	3,090,566–3,980,130
	7,399,253–8,178,263	7,132,660–9,926,208
	37,422,915–38,584,657	
14	3,894,739–4,055,262	
17	2,578,048–2,795,460	
	48,186,828–51,326,457	
	51,667,891–51,907,562	
22	342,310–5,456,059	707,849–3,176,744
23		2,398,565–3,839,815
24		23,382,682–23,398,090
		24,656,717–25,686,788
25	2,091,732–4,600,230	2,605,150–4,576,087
30	972,855–2,482,138	314,526–2,267,036
31	230,277–1,242,356	
	1,641,111–1,805,149	
34	666,725–2,003,315	938,929–1,911,095

## Data Availability

Genomic data can be downloaded from the Gene Expression Omnibus database under accession number GSE121027 (https://www.ncbi.nlm.nih.gov/geo/query/acc.cgi?acc=GSE121027, accessed on 10 April 2022).

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
