# Peer review of "The Shepherd and the Hunter: A Genomic Comparison of Italian Dog Breeds"

_animals, 2023, doi:10.3390/ani13152438_

Round 1

Reviewer 1 Report

The authors of the paper "The Shepherd and the Hunter: A Genomic Comparison of Italian Dog Breeds" performed a genetic comparison between 116 italian hunting dogs (6 breeds) and 158 italian shepherd dogs (9 breeds) using SNP chip data, and FsT, XP-EHH and ROH methods. The main objective is to study the genetic background of these breeds and present their phylogenic relationships.

For me, this paper appears as a follow/complement of a previous paper that the same authors published in this journal this year : Cortellari et al. Genomic Analysis of the Endangered Fonni’s Dog 431 Breed: A Comparison of Genomic and Phenotypic Evaluation Scores. Animals 2023, 13, 818, doi:10.3390/ani13050818. Methods are similar in both papers but here, they make the comparison in terms of the "function/speciality" of each breed.

Since a part of the data / analyses are closed to your previous paper, the comparison with this previous paper is logical. In this context, I have concerns that authors could easily answer to improve the paper.

First, if possible, it will be nice to see picture (in one figure or supp figure) of each breed to help readers to see what your Italian breeds look like. To complete this, a table with breed standards (i.e: height, weight, color, fur, tail...) should be given. Are herding breeds smaller than hunting dog breeds ? This table will help readers to understand why you find LCORL gene for example and will be a good base to do your breed comparison in this present work (L297-304).

Regarding the MDS or phylogenic tree: it is common to include outlier breeds (example : GSD or ESET) or group (ex: molossers) to "root" your analysis. As you cited it in Methods, it seems you have used the same dataset than your previous paper. In this previous paper, you used such breeds (GSD, ESET) to perform the same analysis. Can you do the same here ? Since these breeds are not originally from Italy, they can be considered as outliers in your "italian story". It will also help readers not familiar with such analysis to directly understand where are the hunting dog cluster and the guardian / herding cluster using famous dog breeds. Also, to help readers not familiar with genetic tree, a rectangular tree layout could be more visual/informative than a radial one. Also, can you add value on it (Does the length of the branches mean something?any bootstrap?) 

Since you say your hunting and herding Italian breeds have a common cradle, another way to detect genes associated to hunting or herding capacity could be to perform a GWAS using Plink (--assoc) doing herding vs hunting. Did you try it?

About your breed comparison, it is the weakest part : RSPO2 gene is expected since your are comparing two breeds with different hair texture. So the citation of this example is easy. But you did not discuss about the other new genes you found?  For example, what about SNX29 ? you have a strong signal on this gene when you compare BRAC vs SPIN - A quick search on Pubmed shows that this gene seems to be associated with growth traits and litter size in goat - any information about such phenotype in these dog breeds? any difference in size/weight? you can probably find few new candidate genes in your list to discuss a little bit more, than an easy RSPO2.

L305-312: Very interesting finding - I checked this paper about GDNF and it seems they found a point mutation. Which breeds is driving this result ? Did you check this variant in your collection of dogs ? It could be nice to say you find a new hunting dog breed, probably related to these four breeds described in this paper. I saw you used GSHP and GWHP in your previous paper - maybe you can create a tree with these breeds and find that they are closed to your italian breed(s?) that drive this result.

Minor changes :

In your previous paper, you used the Canine 230K SNP BeadChips (Illumina ®, San Diego, CA, USA). Since you have exactly the same number of dogs for each breed you have tested in both paper, I assume it is the exact same dataset (also as mentioned by your citations in Methods). So, which array did you really used ? 170k or 230k?

L26 : Can you write "encompassing 116 hunting dogs (6 breeds)....and 158 shepherd dogs (9 breeds)."

L87-94: Can you remove "n." before each number ? It is not common to write number like this (or write "n=").

In general, can you fix XPEHH by XP-EHH ?

I am not an english native speaker but I noticed few changes to do :

L190 : it better to write : "350 SNPs were in the top...."

thousand number in english take comma : 15,000 ; 10,000 (L41) ; 3,000 4,000 (L47); 170,000 (L88), 120,828 (L141).

Reviewer 2 Report

1) Simple summary and Abstract gives a different number of investigated breeds (15) than Materials and Methods (16 breeds)

Line 13,14 -  116 hunting dogs (representing 6 breeds) and 158 shepherd dogs (representing 9 breeds). 

15 breeds in total. 

Abstract - line 26 (6 breeds) and line 27 (9 breeds). 

15 breeds in total. 

Materials and Methods - line 85,86  - 274 dogs belonging to 16 different breeds. 

2) Introduction - line 49-52 

In my opinion an important group of dogs of the Roman period should be mentioned, namely Canis Pugnax.

3) Materials and Methods - line 85-94 

In my opinion it would be advisable to analyse at least 20 dogs of each breed. It should be explained why fewer dogs were analysed in the breeds Bracco Italiano (12 dogs), SIPF (16 dogs), SIPR (16 dogs), BERG (15 dogs), DORO (15 dogs), MANN (12 dogs), PALA (10 dogs), SILA (14 dogs). 

Round 2

Reviewer 1 Report

Authors have replied all questions I asked and they now provide all missing informations  in supplementary files (particularly pictures of dogs, very interesting to see). The manuscript is now ready for publications